# Pathological Studies on Hantaan Virus-Infected Mice Simulating Severe Hemorrhagic Fever with Renal Syndrome

**DOI:** 10.3390/v14102247

**Published:** 2022-10-13

**Authors:** Zhouoxing Wei, Kenta Shimizu, Rakiiya S. Sarii, Devinda S. Muthusinghe, Sithumini M. W. Lokupathirage, Junko Nio-Kobayashi, Kumiko Yoshimatsu

**Affiliations:** 1Graduate School of Infectious Diseases, Hokkaido University, Sapporo 060-0815, Japan; 2Institute for Genetic Medicine, Hokkaido University, Sapporo 060-0815, Japan; 3Department of Infectious Diseases and Host Defense, Graduate School of Medicine, Gunma University, Maebashi 371-8511, Japan; 4WISE Program for One Health Frontier Graduate School of Excellence, Graduate School of Veterinary Medicine, Hokkaido University, Sapporo 060-0818, Japan; 5Laboratory of Histology and Cytology, Department of Anatomy, Graduate School of Medicine, Hokkaido University, Sapporo 060-8638, Japan

**Keywords:** Hantaan virus, virulence, mouse model, renal function, neutrophil activation

## Abstract

Hantaan virus is the causative agent of hemorrhagic fever with renal syndrome (HFRS). The Hantaan virus strain, Korean hemorrhagic fever virus clone-5 (KHF5), causes weight loss and renal hemorrhage in laboratory mice. Clone-4 (KHF4), which has a single E417K amino acid change in its glycoprotein, is an avirulent variant. In this study, KHF4 and KHF5 were compared to evaluate pathological differences in mice in vitro and in vivo. The characteristics of the two glycoproteins were not significantly different in vitro. However, the virulent KHF5 strain targeted the lungs and caused pneumonia and edema in vivo. Both strains induced high infectivity levels in the liver and caused hepatitis; however, petechial hemorrhage and glycogen storage reduction were observed in KHF5-infected mice alone. Renal hemorrhage was observed using viral antigens in the tubular region of KHF5-infected mice. In addition, an increase in white blood cell levels and neutrophilia were found in KHF5-infected mice. Microarray analysis of liver cells showed that CD8+ T cell activation, acute-phase protein production, and neutrophil activation was induced by KHF5 infection. KHF5 infectivity was significantly increased in vivo and the histological and clinicopathological findings were similar to those in patients with HFRS.

## 1. Introduction

Orthohantaviruses are enveloped single-stranded negative-sense RNA viruses that belong to the family *Hantaviridae* and the order *Bunyavirales* [1]. Orthohantaviruses are responsible for two fatal rodent-borne zoonotic diseases in humans: hemorrhagic fever with renal syndrome (HFRS) [2] and hantavirus pulmonary syndrome (HPS) [3]. HFRS is caused by Old World hantaviruses, such as the Hantaan virus (HTNV) [2], Seoul virus [4], Dobrava Belgrade virus [5], and Puumala virus [6]. Among the above viruses, millions of HTNV infections have been reported over the past 50 years and it is considered one of the most important viruses in East Asia, and its specific anti-viral agents have not yet been established. Inactivated hantavirus vaccines were developed in China and Korea and clinical trials of DNA vaccines are underway in the United States [7,8,9]. However, most countries do not have a licensed vaccine.

The mechanism underlying HFRS pathogenesis has not yet been elucidated. VE-cadherin cell-signaling is one of the candidates associated with vascular hyperpermeability [10], and reportedly, bradykinin plays a major role in microvascular hyperpermeability [11]. The use of the bradykinin antagonist icatibant, led to the successful treatment of a patient with severe HFRS caused by Puumala virus infection in Europe, supporting the report by Taylor et al. [12]. Inflammatory cytokines are also considered to be associated with pathogenesis of hantavirus [13,14,15]. Interleukin-6 (IL-6) and other inflammatory cytokines are the key to cytokine release syndrome/cytokine storm, causing further inflammatory cytokine outburst [16].

Animal models are important for analyzing viral pathogenicity and planning anti-viral strategies. Among them, laboratory mice are useful experimental animals as a lot of background information and research tools are available. However, laboratory mice rarely exhibit symptoms after inoculation with hantaviruses. Neonatal and severely immunodeficient mice often show symptoms such as encephalitis, pneumonia, and wasting syndrome, which are not typical HFRS symptoms [17,18,19]. Recently, *Nlrc3* knockout (KO) mice infected with HTNV exhibited innate immune disorders, with symptoms similar to HFRS [20]. Therefore, HFRS models using immunocompetent mice are required for the evaluation of anti-viral agents and vaccines.

Previously, we reported that the HTNV strain Korean hemorrhagic fever (KHFV) causes weight loss and renal medullary hemorrhage in mice and can be used as a model for HFRS [21]. CD8+ T cells are involved in renal hemorrhage in this model [22]. We showed that a single amino acid substitution E417K is present in the envelope glycoprotein (GP) in the avirulent clone-4 (KHF4) compared to KHFV [21]. Although this amino acid exchange was not localized within T-cell epitope, a T-cell response was observed, even with KHF4 infection. Thus, the difference in response between the virulent clone-5 (KHF5) and avirulent strain KHF4 in mice is not understood yet.

In this study, we aimed to clarify the viral distribution in mice infected with KHF4 and KHF5. By clarifying the mechanism of HFRS-like symptoms, we sought to demonstrate their suitability as an HFRS model by the comparison of biological markers. Further, we have evaluated the role of the E417K mutation in envelope GPs.

## 2. Materials and Methods

### 2.1. Viruses and Cells

The HTNV strain KHFV was isolated from the blood of a patient with Korean hemorrhagic fever and passaged 10 times in the brains of newborn jcl:ICR mice [23]. Five clones of KHFV were obtained by plaque purification of Vero E6 cells (ATCC CRL1586) [21,23]. Among them, two clones, KHF4 and KHF5, were used in this study. The HTNV strain 76–118 derived from *Apodemus agrarius* was used as a prototype reference [2]. The HTNV strains were propagated in Vero E6 cells maintained in Eagle’s minimum essential medium (EMEM) (Gibco; Thermo Fisher Scientific, Carlsbad, CA, USA) supplemented with 5% heat-inactivated fetal bovine serum (FBS; Biowest, Nuaillé, France), 1% minimum essential medium (MEM) non-essential amino acids (Gibco; Thermo Fisher Scientific), 1% insulin-transferrin-selenium (Fujifilm Wako, Osaka, Japan), 1% penicillin (50 units/mL), streptomycin (50 μg/mL; Nacalai Tesque, Kyoto, Japan), and 1% gentamicin (100 μg/mL; Nacalai Tesque). Virus infectivity titers were determined as described previously [23].

U937 cells (JCRB9021) were purchased from the JCRB Cell Bank and maintained in RPMI 1640 medium (Thermo Fisher Scientific) with 10% fetal calf serum (FCS) (Biowest, Nuaillé, France) and 1% penicillin-streptomycin (Nacalai Tesque). The human adenocarcinoma cell line A549, human embryonic kidney cells, and 293T cells (Riken, Japan) were maintained in Dulbecco’s modified Eagle medium (DMEM) (Thermo Fisher Scientific) with 10% FCS and 1% penicillin-streptomycin.

Splenocytes from mice were prepared as described previously [24]. Splenocytes were maintained in RPMI 1640 medium with 10% FCS, 50 μM 2-mercaptoethanol (Fujifilm Wako), 1% penicillin-streptomycin, and 1% gentamicin (Nacalai Tesque). Dendritic cells were separated from splenocytes using the CD11c Microbeads ultrapure kit (Miltenyi Biotec, Bergisch Gladbach, Germany) as per the manufacturer’s protocols. Peritoneal cells were collected, and adherent cells were used as the peritoneal macrophage fraction.

### 2.2. Indirect Immunofluorescence Assay (IFA)

Vero E6 cells were fixed with acetone one-day post inoculation (dpi) with KHF4 and KHF5. Vero E6 cells expressing the recombinant envelope glycoproteins (rGP) for KHF4 and KHF5 were fixed with acetone two days after transfection (described below). Hybridoma culture supernatants with the dilutions 1:1, 1:10, and 1:100, and secreting monoclonal antibodies targeting HTNV GPs [25] were used. Alexa Fluor 488-conjugated goat anti-mouse IgG (Invitrogen, Thermo Fisher Scientific, A-11029) was used as a secondary antibody at 1:1000 dilution.

### 2.3. Cell Fusion Assay

The cell fusion assay was performed as described previously with a few modifications [26]. The KHFV-inoculated and rGP-expressing Vero E6 cells (see next section) were treated with prewarmed (37 °C) acetate buffered saline adjusted to pH 5.8 for 2 min. The medium was subsequently replaced with growth medium and incubated at 37 °C for 16 h. The cells were washed with PBS, fixed with 10% formalin in PBS, and stained with Giemsa (Merck, Darmstadt, Germany).

### 2.4. Comparison of KHF4 and KHF5 Growth In Vero E6 Cells

Viruses were inoculated into Vero E6 cell monolayers in 6-well plates at a multiplicity of infection (MOI) = 0.1, and the medium was replaced with fresh medium after 1 h of incubation. Subsequently, 150 μL of culture supernatants was collected at 5 dpi, and the viral RNA load was examined by real-time RT-PCR, as described in Section 2.9.

### 2.5. Expression of rGP and Pseudotype Virus Production

The open reading frames for the KHF4 and KHF5 GPs were amplified from viral complementary DNA (cDNA) using the following primer pairs: KHFV_M41F_EcoRI (5′-ATCGAATTCATGGGGATATGGAAGTGGCTAGTGATG-3′) and KHFV_M2450R (5′-TCCTGCTATACCTTATTGTGATG-3′), and KHFV_2250F (5′-GTGCTTGTACAAAGTATGAATAACC-3′) and KHFV_M_M3448R_XhoI (5′-AGCCTCGAGCTATGACTTTTTATGCTTCTTTACGG-3′). The entire open reading frame was connected by overlapping PCR and inserted into the *Eco*RI and *Xho*I sites of the mammalian expression vector pCAGGS/MCS. The sequences of the final plasmid constructs pCAG-KHF4M and pCAG-KHF5M were confirmed by Sanger sequencing. The plasmid vectors were transfected into Vero E6 cells to analyze their antigenicity. The plasmids were also transfected into 293T cells to propagate the pseudotype vesicular stomatitis virus (VSV) coated with recombinant GPs of KHF4 and KHF5, as described previously [27]. The titers of the pseudotype viruses VSV∆G*-KHF4 and VSV∆G*-KHF5 were counted as the number of GFP signals in Vero E6 cells.

### 2.6. Western Blot

Cell monolayers in 6-well plates were inoculated with KHF4 and KHF5 at MOI = 0.1. Five days after inoculation, the cells were lysed with 200 μL SDS sample buffer and heated to 100 °C for 10 min. The lysate (15 μL) was loaded onto an SDS-PAGE gel (e-PAGEL 1020 L, Atto, Tokyo, Japan) and electroblotted onto a 0.45 μm pore immunoblot PVDF membrane (Millipore, Billerica, MA, USA). The N protein of HTNV was detected using the rabbit polyclonal antibody targeting the recombinant N protein of HTNV [28], and a horse radish peroxidase (HRP)-conjugated goat anti-rabbit IgG antibody (Jackson Immuno Research Laboratories Inc., Baltimore, MD, USA, 111-035-003). Bound antibodies were reacted with Amersham ECL Prime (Cytiva, Tokyo, Japan) and detected using an ImageQuant LAS 4000 mini (Cytiva).

### 2.7. Animal Experimentation

Five-week-old female BALB/cCrSlc mice (SLC, Hamamatsu, Japan) were injected with 10^5^ focus forming units (FFU) of KHF4 or KHF5 via the tail vein. The outline of animal experimentation is shown in Appendix A. The weights of the mice were measured from 1 to 14 dpi. At 1, 3, 5, and 7 dpi, two animals were euthanized and blood was collected. If available, urine samples were collected. The lungs, kidneys, spleens, and livers were collected and stored at −80 °C for RNA extraction and viral protein detection. Tissue pieces from the lungs were used for further cultivation to analyze viral replication. Mouse organs were fixed in 4% phosphate-buffered paraformaldehyde (Fujifilm Wako) for histological analysis. All animal experiments were approved by the Animal Studies Ethics Committee of Hokkaido University (19-0088). The mice were treated according to the laboratory animal control guidelines of the Hokkaido University Institutional Animal Care and Use Committee. Experiments involving viral infections were performed at a BSL-3 facility.

### 2.8. Viral Replication in Lung Tissues In Vitro

Lungs were obtained from mice and cut into four pieces. The lung pieces were cultivated in the upper chamber of the culture insert (1 μm pore size, Corning 35310, New York, NY, USA) in 24-well plates using DMEM supplemented with 10% FCS and antibiotics. Briefly, 10^6^ FFU of the KHF4 and KHF5 viruses were inoculated into the lung pieces which were subsequently incubated in 5% CO_2_ at 37 °C. At 24 h after inoculation, the medium was replaced with fresh medium. After 7 days of inoculation, culture medium from the lower chamber was collected, and viral load was analyzed by real-time RT-PCR as described in Section 2.9. Next, lung tissues were obtained from KHF4 and KHF5 administered mice and were cultivated as described. After 7 days of incubation, the culture medium from the lower chamber was collected, and the viral load was analyzed as described.

### 2.9. RNA Extraction and Real-Time RT-PCR

Total RNA was extracted from each tissue using ISOGEN (Nippon Gene, Tokyo, Japan), as per the manufacturer’s protocol. Real-time RT-PCR was performed using the iTaq Universal SYBR Green one-step kit (Bio-Rad Laboratories, Richmond, CA, USA). The qPCR primer sets used were KHFV-SF (5′-TGGACCAAAGGATTATTGTGC-3′) and KHFV-SR (5′-CATCCCCTAAGTGGAAGTTGTC-3′). The mouse β2-microglobulin forward (5′-ACAGTTCCACCCGCCTCACATT-3′) and reverse (5′-TAGAAAGACCAGTCCTTGCTGAAG-3′) primer pair were used to measure the standard gene to compare cell numbers.

### 2.10. Viral Antigen Production in Organs

Approximately 10 mg of tissue was lysed in 200 μL SDS sample buffer and heated to 100 °C for 10 min. The lysate (15 μL) was loaded onto an SDS-PAGE gel and viral N protein was detected by Western blotting as described in Section 2.6. HRP-conjugated anti-GAPDH antibody (Proteintech, Rosemont, IL, USA, HRP-60004) and anti-β-actin antibody (Sigma-Aldrich, A5316, Merck) was used to adjust the total protein concentration in the lysate.

### 2.11. Histopathology and IHC

Tissues were fixed in 4% paraformaldehyde phosphate buffer solution (Fujifilm-Wako) at 4 °C, embedded in paraffin within 14 days, sectioned, and stained with hematoxylin and eosin (HE) (Sapporo General Pathology Laboratory, Sapporo, Japan) to detect the N antigen of KHFV, using the biotinylated E5/G6 monoclonal antibody [29]. Immunohistochemistry (IHC) was performed using the Vector M.O.M. immunodetection kit (Vector Laboratories, Burlingame, CA, USA) following the manufacturer’s protocols.

### 2.12. Microarray Analysis

KHF4, KHF5, and mock-infected mouse livers were collected and stored in RNA Later (Invitrogen). RNA was extracted as described previously, and gene transcription in the liver was analyzed via microarray analysis, performed at GeneticLab Co. Ltd. (Sapporo, Japan). Briefly, extracted RNA was treated using a GeneChip WT Pico Reagent kit (Thermo Fisher Scientific, V.A Graiciuno, Vilnius, Lithuania) according to the manual instructions and then hybridization was performed using Clariom S Array for mice (Applied Biosystems). Data were analyzed using TAC (Thermo Fisher Scientific).

### 2.13. White Blood Cell Population

A blood smear was prepared to examine the white blood cell (WBC) population. After fixing with methanol for 10 min, slides were stained with Giemsa solution. A total of 400 white blood cells were counted on each slide, and the numbers of monocytes, neutrophils, and lymphocytes were recorded. To count the white blood cell number, 2 μL of blood was mixed with Turk stain solution (Nacalai Tesque), and the WBC numbers were counted under a microscope.

### 2.14. Clinical Pathology of Infected Mice

Serum alanine transaminase activity was analyzed as an indicator of liver dysfunction using a serum alanine transaminase (ALT) colorimetric activity assay kit (Cayman Chemical, USA). Blood urea nitrogen (BUN) levels were analyzed as an indicator of renal dysfunction (DetectX Urea Nitrogen Colorimetric Detection Kit, Funakoshi, Tokyo, Japan). Urinary proteins were analyzed by SDS-PAGE, followed by protein staining (Simply Blue, Invitrogen, Thermo Fisher Scientific). IL-6 and tumor necrosis factor (TNF)-α levels were evaluated to detect the systemic cytokine storm using a mouse IL-6 detection kit (Chondrex #6702, Woodinville, WA 98072, USA) and mouse TNF-α detection kit (Chondrex #6701), respectively, following the manufacturer’s protocols.

### 2.15. Statistics

The paired student’s *t*-test was used, and statistical significance was set at *p* < 0.05. Significance was assigned as * *p* < 0.05, ** *p* < 0.01, and *** *p* < 0.001. Images of Western blot bands yielded by anti-N antigen and GSPCF antibodies were quantified using open-source Fiji/Image-J software (https://imagej.net accessed on 14 September 2022).

## 3. Results

### 3.1. Characterization of KHF4 and KHF5 GPs

The amino acid at position 417 in the GP Gn is the sole difference between the viral proteins of KHF4 and KHF5, and the changed amino acid may be responsible for virulence in mice. However, the role of this substitution has not been clarified. To investigate the role of this mutation, we compared the antigenicity, fusion activity, and growth of the two viruses and their recombinant GPs. First, antigenic profiling using 23 types of monoclonal antibodies against the HTNV GP indicated no remarkable antigenic difference between the KHF4 and KHF5 GPs (Figure 1A), KHF4 and KHF5-inoculated and rGP-expressing Vero E6 cells were exposed to low pH. As shown in Figure 1B, both KHF4 and KHF5 showed high cell fusion activity, indicating that the single amino acid substitution E417K on the GP has no effect on fusion activity. Next, viral replication was analyzed in Vero E6 cells. KHF4 and KHF5 were inoculated into Vero E6 cells at an MOI of 0.1, and the supernatant was collected from 1 to 10 dpi to determine the amount of viral RNA present using real-time RT-PCR (Figure 1C). The viral RNA copy number of KHF5 was significantly higher than KHF4 since 3 dpi. To compare viral entry, pseudotype VSV coated with the rGPs of KHF4 and KHF5 were generated (Figure 1D). Both pseudotyped viruses showed high and similar titers. This observation indicates that both GPs were assembled into VSV virions without any difference.

### 3.2. The Cell Tropism of KHF4/KHF5 In Vitro

To identify the factors responsible for the virulence change caused by the E417K mutation, the cell and organ tropism of KHF4 and KHF5 were examined in vitro. As shown in Figure 2, the production of viral N protein after inoculation with KHF4 and KHF5 was compared in Vero E6, A549, U937, and HEK 293T cells. KHF4 showed higher N protein production than KHF5 in all cell lines except U937. Vero E6 cells were used for amplification of HTNV strain 76–118. A549 cells were originally used for amplification of HTNV 76–118 [30]. Conversely, U937 cells were used for antibody-dependent enhancement of HTNV 76–118 [31]. Without anti-GP antibodies, U937 cells were not infected with HTNV 76–118. Similarly, KHF4 and KHF5 had no infectivity against U937 cells in the current experiments. In mouse in vitro experiments, KHF4 showed higher N protein production than KHF5 in peritoneal macrophages, dendritic cells, and splenocytes, similar to in the cell lines. Compared with the prototype HTNV strain 76–118, which was not pathogenic in immunocompetent mice, both KHFV strains showed high infectivity toward immune cells.

According to the observations from Vero E6 cells shown in Figure 1C and Figure 2A, KHF4-infected cells kept N protein in cells more than KHF5-infected cells; however, virus release was significantly lower than in KHF5-infected cells. The tendency of intracellular N protein to be higher in KHF4-infected cells is consistent in various cells has shown in Figure 2. The results suggest slight differences between KHF4 and KHF5 in the release of viral progeny. Next, we needed to examine differences between KHF4 and KHF5 in mice to understand KHF5 pathogenesis.

### 3.3. KHF5 Targeting the Lungs May Play an Essential Role in Pathogenesis

As reported previously, BALB/c mice infected with KHF5 showed weight loss at 5 dpi (Figure 3A) [21], and severe kidney hemorrhage occurred at 7 dpi (Figure 3B), from which the mice subsequently clinically recovered, based on symptoms such as ruffled fur and decreasing activity. As shown in Figure 4A, the viral RNA loads in the lung, liver, spleen, and kidney were determined at 1,3, 5, and 7 dpi using real-time RT-PCR. The viral RNA load in the lung and kidneys was significantly higher in KHF5 than in KHF4-inoculated mice at 3 dpi. In the lungs, high levels of viral N protein were detected in mice with KHF5 infection alone (Figure 4B). The results suggest that the lungs are an important site for viral replication during the early phase of infection in mice. A high viral load in the liver was detected in both the KHF4 and KHF5-inoculated mice. Significantly higher viral loads were observed in KHF5-inoculated mice at 5 and 7 dpi. Viral N protein was detected in the liver, and the ratio of protein was almost the same as that in the lung when compared with the reference protein GAPDH in KHF5-infected mice (Appendix A). Conversely, in KHF4-infected mice, N protein production in lungs was lower than in KHF5-infected mice. The results suggest that the lungs are an important site for viral replication of KHF5. Viral N protein in spleens were below detectable levels.

### 3.4. Viral Replication in the Lung Tissue

To confirm the KHF5 tropism in the lungs, lung tissues from infected mice were cultured. After 7 days, culture supernatants were collected from the lower chamber and viral load was examined by RT-PCR. Viral RNA load in the medium were higher for KHF5 than KHF4 (Figure 5A). Next, KHF4 and KHF5 were inoculated into the lung tissue obtained from mice with no viral inoculation, and the tissues were subsequently cultured for 7 days (Figure 5B). KHF4 showed higher viral production in the culture supernatant and N protein production in the lung tissue than KHF5 (Figure 5C).

### 3.5. KHF5 Infection Caused Renal Hemorrhage, Lung Edema, and Acute Hepatitis

As reported previously [21], KHF5 infection causes renal hemorrhage. As KHFV targeted the lungs and both strains showed high infectivity to the liver, viral N antigens were analyzed by IHC to examine the distribution of viruses in the lung, liver, and kidneys. In the KHF5-infected lungs of mice, viral antigens were detected in alveoli and bronchial epithelial cells, compared with the negative control and KHF4-infected lungs. KHF5 infection caused severe pneumonia, and aggregation of lymphoid cells was identified near the peri-bronchus edema. Additionally, mononuclear cell infiltration with KHFV antigens was found in capillary vascular endothelial cells (Figure 6A). As both strains showed high infectivity in the liver, the viruses caused acute hepatitis due to lymphocytic infiltration of inflammatory cells. Reduced glycogen storage and petechial hemorrhage were observed in KHF5-infected mice alone (Figure 6B). Focal interstitial hemorrhage was observed in the renal medulla, cortex, and tubular region in KHF5-infected mice alone, and viral antigens were detected mainly in the tubular region (Figure 6C).

### 3.6. KHF5 Infection Caused Acute Hepatitis and Neutrophilia

Acute hepatitis with lymphocytic infiltration of immune cells was observed in the KHF4 and KHF5 sections; therefore, liver dysfunction was assessed via ALT activity analysis. As shown in Figure 7A, serum ALT activity was significantly increased in both KHF4 and KHF5-infected mice at 7 dpi. In contrast, BUN levels did not change at 5 dpi and 7 dpi despite renal hemorrhage (Figure 7B). We collected blood from each mouse at 1, 3, 5, and 7 dpi to examine white blood cell population levels, which showed a decrease in white cell numbers at 5 dpi, and an extreme increase at 7 dpi. At 5 dpi, KHF5-infected mice showed a high ratio of neutrophils, indicating that neutrophilia may be related to pathogenesis in this phase (Figure 7C). Serum IL-6 and TNF-α were examined but both of them were not detected (Appendix A). To analyze renal damage, we examined renal proteins by SDS-PAGE (Appendix A). Compared to urine from control mice, multiple bands were evident in the urine of infected mice, indicating that functional changes may have occurred. However, albuminuria was not detected. Notably, 120 KDa bands were observed in KHF5-inoculated mice at 7 dpi, though the protein sequences could not be identified (data not shown). Urine kidney injury molecule (KIM-1) levels were also examined using a KIM-1 detection kit, but no increase in KIM-1 levels was detected (data not shown).

### 3.7. KHFV Infection Induces T Cell and Neutrophil Infiltration in the Liver

We found that KHF4 and KHF5 show high levels of replication in the liver and cause hepatitis. To analyze the pathogenicity of these viruses in the liver, mRNA expression was examined using microarray analysis. A total of 20,226 genes showing transcriptional changes were compared among the KHF4, KHF5, and mock infection groups. As shown in Appendix A, the expression of several genes was altered between the infected and control groups. However, few genes differed in expression between the KHF4 and KHF5 groups. Increased expression of genes in CD8+ T cells and that of licocalin-2 was identified in both KHF4 and KHF5-infected liver cells. As shown in Figure 8, the levels of cytotoxic T cell (granzyme A and B) and neutrophil-related genes (lipocalin-2/neutrophil gelatinase-associated lipocalin) were increased. In KHF4-infected mice, the expression of CD8+ cell-related genes showed a greater increase than in KHF5-infected mice. In contrast, lipocalin-2 expression levels were changed in KHF5-infected mice more than in mice infected with KHF4. These genes were derived from cells that infiltrated into the liver from the blood or hepatocytes. In contrast, acute phase proteins produced by hepatocytes (serum amyloid A2 and A3) were higher in both KHF4 and KHF5-infected mice, indicating that viruses induce inflammation in the liver, but KHF5 may cause a predominant response to neutrophils.

## 4. Discussion

In this study, KHF5 was used to investigate HFRS-like pathogenesis in mice. We previously reported that the amount of viral RNA in the kidneys of KHF5-infected mice was greater than that in KHF4-infected mice and peaked before the onset of disease [21]. However, we had not fully examined organ tropism and histological data to evaluate whether this experimental mouse was suitable as a model for human HFRS. Therefore, we analyzed the renal pathology in this study. The viral antigen and petechial hemorrhage were identified in the tubular region, which induced HFRS symptoms in humans [32].

In addition, we found that the pathogenic strain KHF5 alone targeted the lungs and caused lung edema in the HFRS mouse model. The lungs may be considered a systemic source of the virus. We also found neutrophil infiltration near the bronchus edema. Previously, we reported that the depletion of neutrophils can suppress pulmonary vascular hyperpermeability and the occurrence of pulmonary edema in SCID mice [19]. In humans, a few HFRS patients reportedly develop respiratory disorders [33,34]. Inhalation of excreta from wild rodents is considered the main route of hantavirus transmission to humans, and an epidemiological study on the Puumala virus (PUUV), one of the causative agents of HFRS in Europe, indicated a relationship between smoking and seropositivity [35]. Burdening the lungs by smoking can increase the efficiency of hantavirus infection. These reports suggest that the lungs play an essential role as the primary target site in the initial phase of hantavirus infection.

Interestingly, we found that both strains showed extremely high infectivity towards the liver, and acute hepatitis and lymphatic infiltration of inflammatory cells were observed in KHF4 and KHF5-infected mice. ALT levels are used as indicators of liver disease. Based on the results, we found that KHF viruses cause acute hepatitis in mice. Serum ALT activity and liver section analysis showed that KHF4 and KHF5 infections caused hepatitis; however, glycogen storage reduction and petechial hemorrhage in the presence of viral antigens were detected in mice with KHF5 infection. Significant increases in ALT activity have been frequently reported in HFRS cases, and liver damage is considered important for hantavirus pathology [36,37].

We counted the number of white blood cells in mice at 1, 3, 5, and 7 dpi, and found that leukocyte levels decreased from 3 to 5 dpi and were subsequently increased at 7 dpi, which was similar to the pattern observed in two patients with severe HFRS infection as reported previously [38]. In addition, Strandin and colleagues reported the presence of neutrophil activation in acute HFRS [39]. A similar increase in neutrophils was observed in KHF5-infected mice at 5 dpi, the onset points of weight reduction and renal hemorrhage, and the levels were subsequently reduced at 7 dpi at a severe stage of disease which was considered to indicate the beginning of recovery. The change in neutrophil levels followed by the change in symptoms indicates the importance of neutrophils in HFRS pathogenesis.

Cytokine storms may play a crucial role in the manifestation of both HFRS and HPS [13,15]. To examine the cytokine storm in mice, the levels of IL-6 and TNF-α in the blood at 7 dpi were analyzed, but no significant differences were found among the KHF4, KHF5, and negative control groups. Therefore, a systemic cytokine storm was not considered to have occurred in these mice.

The microarray analysis results showed that both KHF4 and KHF5 showed high expression levels of granzyme A and granzyme B in the liver, which were produced mainly by CD8+ T cells. Recently, we reported that CD8+ T cells are involved in the development of renal hemorrhage in KHF5-inoculated mice [22]. Notably, non-pathogenic KHF4 induced higher expression of granzyme A and B than KHF5. These results suggest that CD8+ effector T cells may be involved in renal injury and recovery from infection. Lipocalin-2 expression showed a 393-fold and 445-fold increase in KHF4 and KHF5-infected mice, respectively, compared to the negative control group. Lipocalin-2 is mainly expressed by neutrophils and hepatocytes and is induced by IL-6 in mice [40]. In this study, the source of lipocalin-2 in the infiltrated neutrophils or hepatocytes was not determined. However, the increase in lipocalin-2, serum amyloid A2, and A3 gene expression levels in the microarray analysis indicated the induction of inflammatory cytokines in the liver. An increase in BUN levels or albuminuria were not observed in KHF4 and KHF5-inoculated mice, despite the presence of renal hemorrhage. It was difficult to collect enough volumes of urine specimens to examine KIM-1 from mice. In HFRS patients, KIM-1 elevation in urea was reported [41]. In the present study, KIM-1 concentrations in a few urine samples were examined; however, we could not detect high concentrations of KIM-1. Further investigations are required to evaluate renal dysfunction in KHF5-infected mice.

Hantavirus is beginning to be recognized as a virus that causes hepatitis [42]. Recently, it was reported that the livers of host rodents are infected with hantaviruses [43]. Seoul orthohantavirus targets the microvasculature in the liver of chronically infected asymptomatic rats [44]. In the present study, KHF5 could infect parenchymal cells of the liver more frequently than endothelial cells. Moreover, antigen-positive hepatocytes were dispersed. However, there are very few reports on hantavirus infectivity to hepatocytes. Once the types of hepatocytes targeted by hantavirus in the liver are known, novel techniques, such as single-cell RNA sequencing, would provide useful information.

As demonstrated in a previous paper, under KHF5 infection, T-cells are obviously involved in the expression of pathogenicity, because weight loss was not observed following CD8+ T cell depletion [21,22]. Results from microarray analysis of liver also suggest T-cell activation. Conversely, virus specific CD8+ T cells are also effective in recovery [24]. Suppression of CD8+ T cells is probably associated with the establishment of persistent infection [45,46]. Since KHF, especially KHF5, replicates faster in mouse tissues than prototype Hantaan virus strain 76–118, T-cell activations tending towards increasing inflammation rather than recovery. 

Although a single amino acid mutation in Gn altered the virulence of KHFV in mice, we did not determine the cause of the difference. Under KHF5 infection, viral RNA was higher in the culture supernatant, whereas viral N protein was reduced in the cells. In contrast, viral N protein showed higher induction in KHF4-inoculated cells in vitro and in lung tissues. The results indicate that avirulent mutations may affect the release of daughter viruses. The minor difference may ultimately result in a distinct pathogenicity difference in mice. Conversely, KHF5 progeny viruses in the lungs were high under in vivo inoculation, but low under lung tissue inoculation, indicating that an unknown factor favors the growth of KHF5 in lungs of mice.

## 5. Conclusions

In this study, KHF5-infected mice showed phenotypes similar to human HFRS cases, such as renal hemorrhage, pneumonia, hepatitis, and other clinicopathological changes. However, the phenotypes confirmed in this study did not include renal dysfunction. In mice, indicators besides BUN and albuminuria should also be considered. Furthermore, inflammatory cytokines were induced but did not cause a cytokine storm in the mouse model, and the pathology progressed along with the activation of neutrophils. These observations suggest that KHF5-infected mice may be useful models for severe HFRS infection.

## Figures and Tables

**Figure 1 viruses-14-02247-f001:**
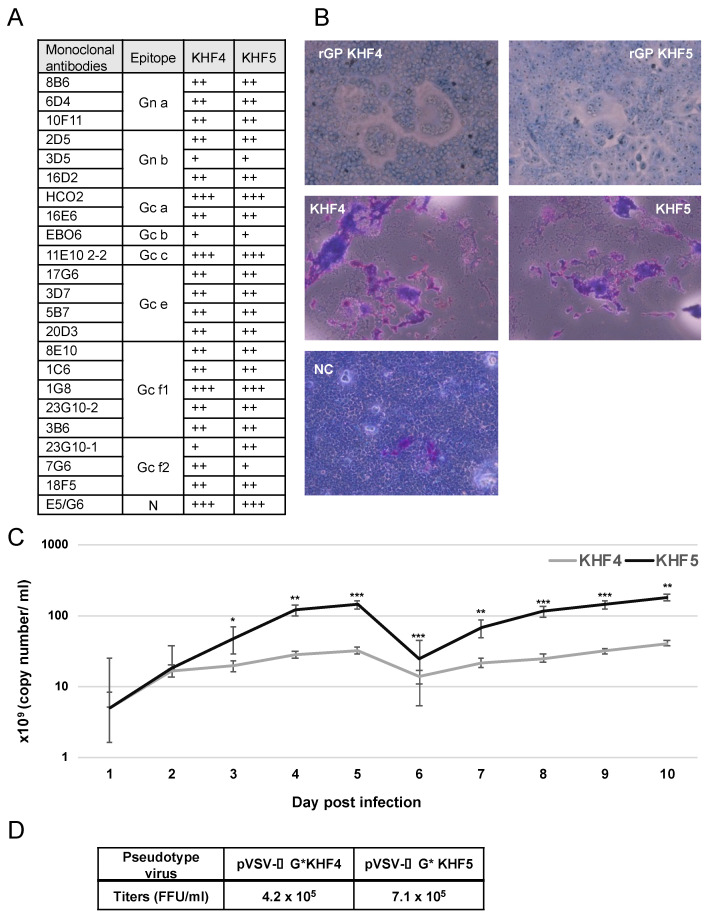
Virological characterization of KHF4 and KHF5. (**A**): GPs of KHF4 and KHF5 were compared for monoclonal antibody binding. The culture supernatants of hybridoma of monoclonal antibodies were diluted 1:1, 1:10, and 1:100, and positivity in the 1:100, 1:10, and 1:1 dilutions was assessed as +++, ++, and +, respectively. (**B**) Cell fusion assay. Cell fusion activity of recombinant glycoproteins of KHF4 and KHF5 expressed in Vero E6 cells by transfection (upper panels). Fusion activity of Vero E6 cells infected with KHF4 and KHF5 (central panels). Cells were inoculated with viruses at a MOI of 0.1 and incubated for 5 days. Uninfected Vero E6 cells were used as negative controls (NC; lower panel). (**C**) Viral growth in Vero E6 cells. Culture supernatant was collected from cells between 1 to 10 dpi, and the viral load was quantified by real-time RT-PCR (Significance was assigned as * *p* < 0.05, ** *p* < 0.01, and *** *p* < 0.001). (**D**) Pseudotype VSV with KHF4 and KHF5 glycoproteins were transfected to HEK293T cells and the propagated pseudotype VSV and titers were determined in Vero E6 cells.

**Figure 2 viruses-14-02247-f002:**
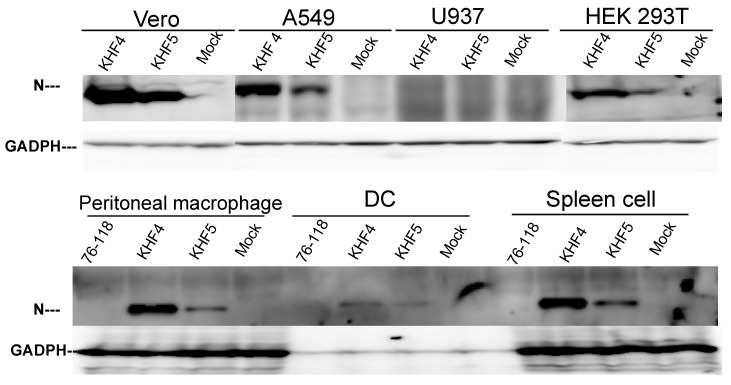
Cell tropism of KHF4 and KHF5 in vitro. Viral N protein production was compared by Western blotting to clarify virus replication in various cells. Four cell lines, (Vero E6, A549, U937, and HEK 293T) and peritoneal macrophages, dendritic cells (DC), and spleen cells obtained from five-week-old BALB/c mice were inoculated at a MOI of 0.1 and cultured for 5 days. GAPDH was used as the reference protein.

**Figure 3 viruses-14-02247-f003:**
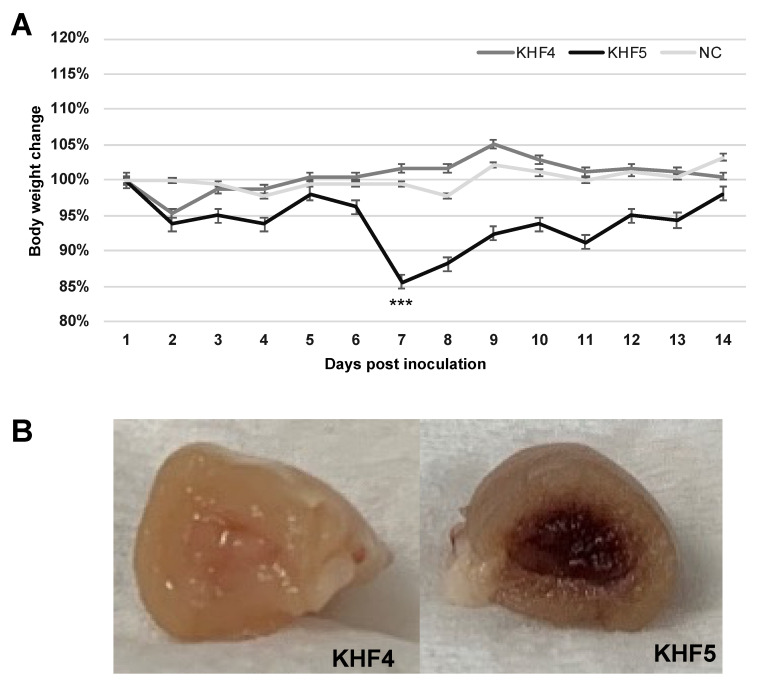
Body weight and renal changes in mice with KHF4 and KHF5 infection. (**A**) Five-week-old BALB/c mice were infected with KHF4 and KHF5 strains, or mock inoculation with culture medium of Veo E6 cells (NC). Subsequently the weight changes were measured between 1 and 14 dpi. *** *p* < 0.001. (**B**) At 7 dpi, a significant decrease in weight loss was observed, indicating renal hemorrhage.

**Figure 4 viruses-14-02247-f004:**
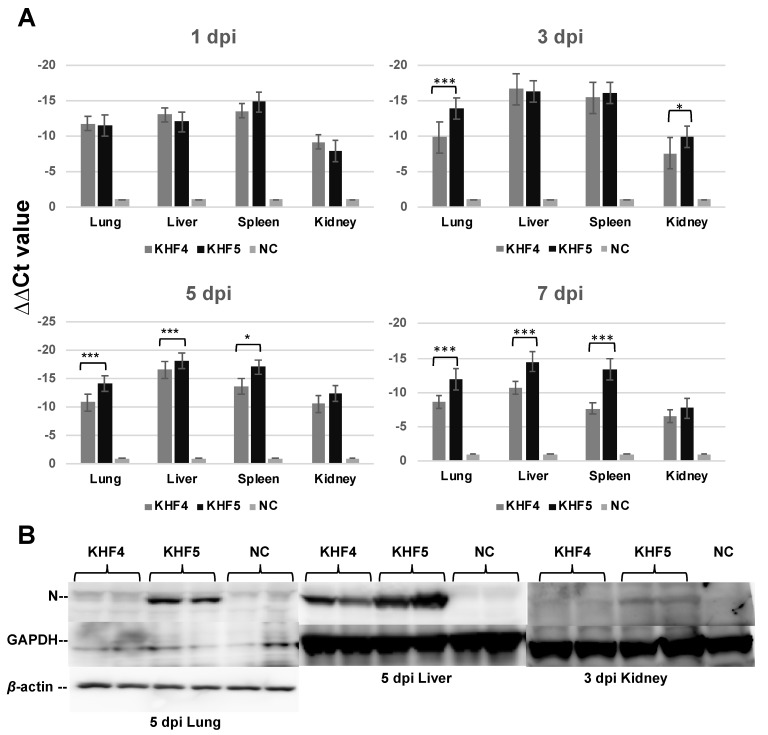
Viral replication in the lungs, liver, spleen, and kidneys. (**A**) Viral load in the four organs were quantified via real-time RT-PCR. The viral load was detected using S segment primer sets, normalized relative to β-2 microglobulin, and compared with the negative control (∆∆Ct KHFV = (Ct _KHFV S_ − Ct _KHFV_
_β2M_) − (Ct _NC S_ − Ct _NC β2M_)) (Significance was assigned as * *p* < 0.05, and *** *p* < 0.001). (**B**) The anti-HTNV viral N protein was detected by Western blotting. Results from each pair of mice are shown.

**Figure 5 viruses-14-02247-f005:**
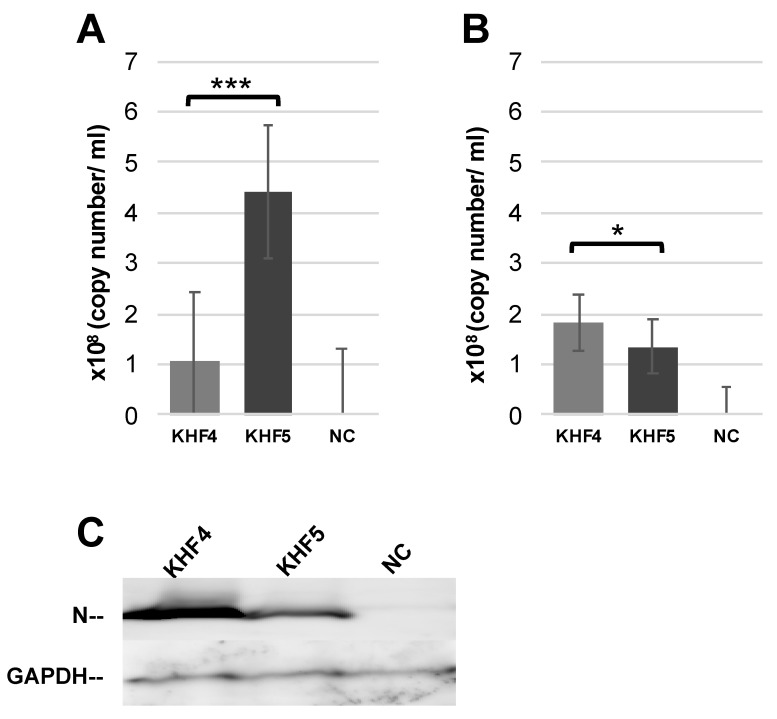
KHF4 and KHF5 replication in lung tissue. (**A**) Lungs were collected from KHFV-infected mice at 5 dpi and cultured. After 7 days, the supernatant was collected for real-time RT-PCR. (**B**) Uninfected BALB/c mouse lungs were collected and cultured as described above. After infection with KHFV, the supernatant was collected at 7 dpi for real-time RT-PCR analysis. (**C**) Viral N protein in the lung tissue as shown in Figure 5B detected by Western blot. Results from mock inoculated mouse and lung tissue are shown in NC (Significance was assigned as * *p* < 0.05, and *** *p* < 0.001).

**Figure 6 viruses-14-02247-f006:**
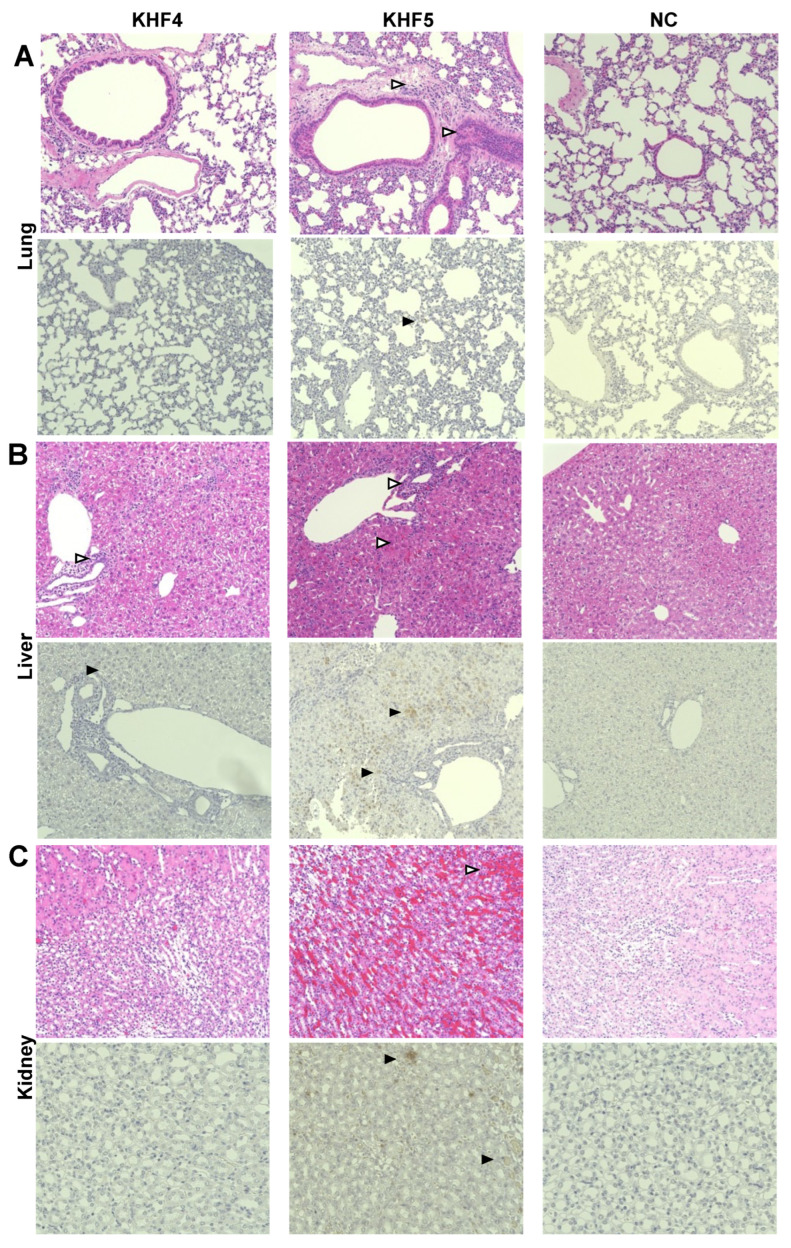
Immunohistochemistry analysis of tissues from KHF4 and KHF5-infected mice. Tissue slices were stained with hematoxylin and eosin (HE), and viral antigens were detected using a biotinylated E5G6 monoclonal antibody with the Vector M.O.M immunodetection kit. Characteristic lesions were indicated by open triangles. Viral antigens are indicated by closed triangles. (**A**) Peri bronchus edema with lymphocyte aggregation was observed in KHF5-infected lungs, and mild antigen reaction was observed in KHF5-infected alveolar epithelial cells. (**B**) Petechial hemorrhage was observed in KHF5-infected livers by HE staining, whereas KHF5-infected livers showed multiple antigenic reactions and hemorrhage. The KHF4-infected liver also showed perivascular lymphoid infiltration. (**C**) Renal hemorrhage caused by KHF5 infection was distributed in the cortex-medullary region, and antigens were detected in the tubular regions.

**Figure 7 viruses-14-02247-f007:**
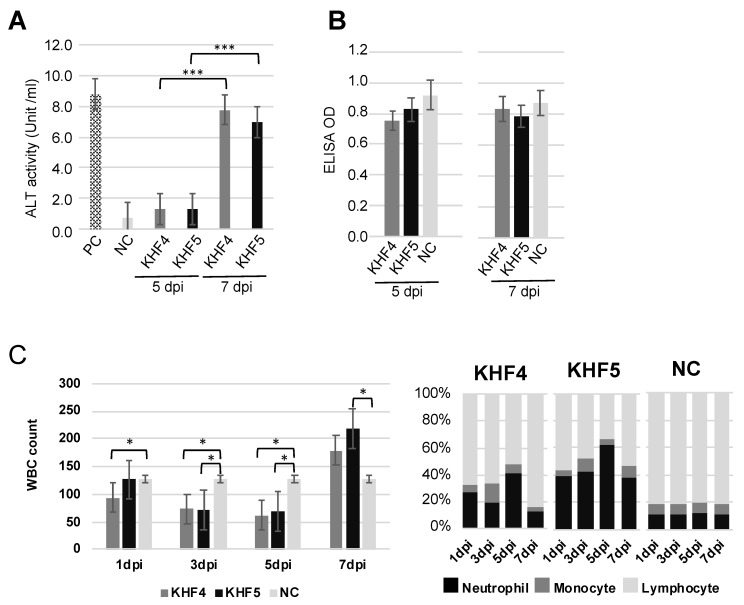
Clinicopathological tests by peripheral blood and serum. (**A**) Fluctuation of ALT for liver dysfunction. (**B**) Fluctuation of BUN. MEM medium injected into five-week-old BALB/c mice was used as the negative control (NC), and the positive control (PC) was provided with an ALT kit. The optical density was read at 340 nm and 450 nm, and ALT activity and BUN analysis were performed as per the manufacturer’s instructions. (**C**) White blood cell number per μL of blood (**left** panel) and in the population (**right** panel) (Significance was assigned as * *p* < 0.05, and *** *p* < 0.001).

**Figure 8 viruses-14-02247-f008:**
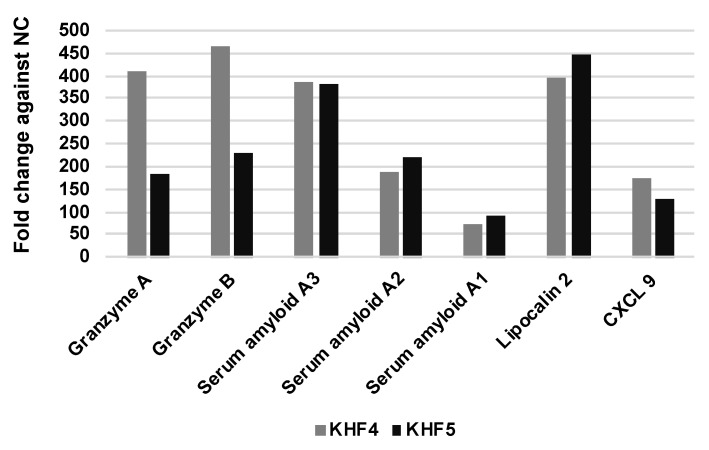
Liver gene expression profiling using microarrays. Transcriptional changes were compared in KHF4, KHF5, and mock-infected mice. The top ten genes showing the maximum fold-change difference in KHF4/KHF5-infected mice relative to the controls were selected and compared.

## Data Availability

The datasets generated during the current study are available from the corresponding author upon reasonable request.

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
