# Peer review of "Pathological Studies on Hantaan Virus-Infected Mice Simulating Severe Hemorrhagic Fever with Renal Syndrome"

_viruses, 2022, doi:10.3390/v14102247_

Round 1
Reviewer 1 Report
The manuscript of Wei and co-workers describes an experimental study to compare the pathogenicity of two different strains (KHF4 and KHF5) of Hantaan virus (HTNV) that differ only by a nucleotide substitution causing an amino acid exchange in the glycoprotein. The pathogenicity of both strains was compared in mice in vitro and in vivo. While the in vitro studies did not show significant differences between both strains, the in vivo study in the mice model indicated large differences. From these results the authors concluded that the KHF5 strain in mice represents a promising disease model for hemorrhagic fever with renal syndrome (HFRS).
Animal disease models that reflect the clinical course of HFRS are still rare, but urgently needed to understand the underlying processes resulting in pathologies, but also to develop and approve antiviral therapies. Therefore, this study is of significant importance for the hantavirus research community, but also for research on other hemorrhagic fever viruses. However, I have several major issues that needs to be addressed by the authors:
1. The authors have reported, as cited, the potential of the KHF strain as a model for HFRS and the involvement of cytotoxic T-cells here (see lines 59-63). The authors should clearly state what is the (additional) rational of the recent study.
2. The terminology should be proven and corrected along the entire manuscript: e.g. what is meant by “aggressively increased” (abstract, line 31); the authors should clearly differentiate between amino acid exchanges such as E417K and nucleotide substitutions (see e.g. line 22 – this is an amino acid exchange, not a mutation). Using a real-time RT-PCR, a RNA load can be determined, not the “viral load” directly (see lines 115/116). Abbreviations need to be introduced (see line 201: WBC – needs to be introduced on line 200). The authors should use “HTNV strains” instead of “HTNVs” (see line 78). What is meant by “systemic viral dynamics” (line 67)? “viral cDNA” (lines 115/116)? “viral fusogenicity” (line 227)? The authors should use “real-time RT-PCR” instead of “real-time PCR”.
3. The description of the Materials and Methods needs to be improved: How many animals were used for the infection experiments? (chapter 2.7). Explain “cCrSlc” mice. Lines 114/115: “after 1h of incubation with the viruses at 5 dpi” – please clarify! The authors should prove the location of the companies – see e.g. line 140.
4. In general, the results of the study are valid and well documented by the figures and tables presented. However, the quality of some figures or the corresponding legends needs to be improved. Fig. 2: Why the HTNV prototype strain 76-118 was used only for cells given below, but not for Vero, A549, U937 and HEK293T cells? Figure 3: the NC used is not given in the legend, what is shown in panel B? Fig. 4B: The GAPDH controls for the samples from lung at 5 dpi (left panel) raise the question if the results for N protein can be compared. What is shown on the left and right lane, e.g. for KHF4? Fig. 5: The quality of panel C is bad and should be substituted by a better one. What is used as NC? Fig. 6: The labelling in the figure differs from that given in the legend, e.g. in the figure itself panel B is labelled as “liver,” but in the legend it is given as “lung”. Same seems to be the case for panel C. The figure itself is hard to interpret – What should be seen at the asterisks? This information should be given in the legend.
5. The authors should prove if their statements or conclusions are really based on the results of their experiments – see lines 235-237. The results described may result to this conclusion, but there is no direct proof for this assumption. Can the authors correlate the gene expression profile with HTNV infection on single cell level (chapter 3.7)? Prove the use of words like “modestly relative” (lines 230/231).
Additional minor points:
Line 42: infections
Lines 41-44: What is about anti-HTNV vaccines in China?
Line 46: “candidate systems”?
Line 52: “mouse” – do you mean “house mouse (Mus musculus) – derived laboratory mice”?
Lines 63/64: this amino acid exchange is not localized within a T-cell epitope
Lines 80, 160, 343: Introduce abbreviations MEM, FFU, KIM
Line 161: correct carbon dioxide (2)
Line 270: Is that shown in Fig. 3B?
Line 284: Viral – RNA load
Figure 7: correct “lymphocyte” in panel C
Line 383: in humans
Line 391: PUUV is ONE of the causative agents of HFRS in Europe, but not the only one.
Line 438: progeny viruses
Lines 468/469: No investigation of human samples! Therefore, this statement is inappropriate.
Reviewer 2 Report
Lines 54-55. Immunocompetent mice are required for what? This can be deleted since it is said again on line 58.
Lines 230-231. What is meant by modestly relative? Similar? How different are they?
Figure 6. When describing the lesions indicate that an arrow points them out.
Lines 415 - 219. How important are cytokine storms in the pathogenesis of human HFRS? Does the absence of cytokine storms in mice detract from the use of mice as a model for human pathogenesis?
Reviewer 3 Report
There are some problems with this paper:
1. This study aims to figure out the systemic viral dynamics in vitro and vivo infected with KHF4 and KHF5. Why choose the KHF4 and KHF5 in this study?
2. In figure 2, an internal reference is required to compare the expression levels of viral N protein between different viral infections. And the figure 2 showed that the expression of KHF4 N protein was lower than the KHF5, which was contradictory to the statement in line 255-256.
3. In figure 3A, there was no weight loss observed in the mice infected with KHF4. Please author check the statement of line 268.
4. In line 270, there was no result to support the conclusion“the mice subsequently recovered”.
5. In figure 4B, the WB did not detect the band of GAPDH in lungs, so the level of viral N protein could not be compared among the different viral infection.
6. The Statistics from three independent WB experiments are required to illustrate the conclusion “Viral N protein was detected in the liver, but the amount of protein was lower than that in the lung compared to the reference protein GAPDH”.
7. The levels of IL-6 and TNF-alpha in the blood at 7 dpi were not shown.
8. In general, the conclusions are overstated, more experimental data should be need to support renal dysfunction of HFRS mice model.
9. It is noted that the manuscript needs careful editing by someone with expertise in English writing.
Round 2
Reviewer 3 Report
1. In figure 2, an internal reference is required to compare the expression levels of viral N protein between different viral infections. And the figure 2 showed that the expression of KHF4 N protein was lower than the KHF5, which was contradictory to the statement in line 255-256.
As you pointed out Western blot images were in low quality and without internal reference. I apologized this mistake. We have carefully replaced these images. On the other hand, upper panels of Fig. 2 were not used GAPDH because we used constant amount of line cells.
The original WB photograph of GAPDH can not be find in the original images file folder. In addition, internal reference is necessary for indicating the constant amount of line cells.
1. In figure 4B, the WB did not detect the band of GAPDH in lungs, so the level of viral N protein could not be compared among the different viral infection.
A better-quality photo was added, due to the lower cell density, the GADPH detected from lung is lower than the other organs.
I agree your suggestion and replaced in all figures with better-quality photos. In revised manuscript, we replaced photo for GAPDH in lung tissues.
The level of GAPDH in lung tissues is too low, which is insufficient as an internal reference.
